# A Mini-Review of Diagnostic Methods for the Antigen and Antibody Detection of Rocky Mountain and Brazilian Spotted Fever

**DOI:** 10.3390/biomedicines12071501

**Published:** 2024-07-06

**Authors:** Kamila Alves Silva, Vanesa Borges do Prado, Rafael Rodrigues Silva, Marcelo van Petten Rocha, Rafael Almeida Ribeiro de Oliveira, Tarumim de Jesus Rodrigues Falcão, Clara Cristina Serpa, Marina Andrade Rocha, Sabrina Paula Pereira, Líria Souza Silva, Juliana Martins Machado, Ricardo Andrez Machado-de-Ávila, Ricardo Toshio Fujiwara, Miguel Angel Chávez-Fumagalli, Eduardo Antônio Ferraz Coelho, Rodolfo Cordeiro Giunchetti, Mariana Campos-da-Paz, Ana Alice Maia Gonçalves, Alexsandro Sobreira Galdino

**Affiliations:** 1Programas de Pós-graduação em Biotecnologia (PPGBIOTEC) e Multicêntrico em Bioquimica e Biologia Molecular (PMBqBM), Disciplina Biotecnologia & Inovações, Universidade Federal de São João Del-Rei, Divinópolis 35501-296, Minas Gerais, Brazil; kamilasilva504@gmail.com (K.A.S.); vanessa.borges97@hotmail.com (V.B.d.P.); rafaelrodrigues.silva@outlook.com (R.R.S.); vanpetten.bqi@gmail.com (M.v.P.R.); rafaaholiveira96@gmail.com (R.A.R.d.O.); falcaotarumim@gmail.com (T.d.J.R.F.); claracserpa@gmail.com (C.C.S.); marinaar99@gmail.com (M.A.R.); sabrina.docbiotec@gmail.com (S.P.P.); 2Laboratório de Biotecnologia de Microrganismos, National Institute of Science and Technology in Industrial Biotechnology (INCT-BIO), Universidade Federal de São João Del-Rei, Divinópolis 35501-296, Minas Gerais, Brazil; liriasza@aluno.ufsj.edu.br (L.S.S.); julianam.m@hotmail.com (J.M.M.); anafish@hotmail.com (A.A.M.G.); 3Laboratório de Fisiopatologia Experimental, Programa de Pós-Graduação em Ciências da Saúde, Universidade do Extremo Sul Catarinense, Criciúma 88806-000, Santa Catarina, Brazil; r_andrez@unesc.net; 4Departamento de Parasitologia, Instituto de Ciências Biológicas, Universidade Federal de Minas Gerais, Belo Horizonte 31270-901, Minas Gerais, Brazil; fujiwara@icb.ufmg.br; 5Computational Biology and Chemistry Research Group, Vicerrectorado de Investigación, Universidad Católica de Santa María, Arequipa 04000, Peru; mchavezf@ucsm.edu.pe; 6Postgraduate Program in Health Sciences: Infectious Diseases and Tropical Medicine, Faculty of Medicine, Federal University of Minas Gerais, Belo Horizonte 30130-100, Minas Gerais, Brazil; eduardoferrazcoelho@yahoo.com.br; 7Laboratory of Biology of Cell Interactions, National Institute of Science and Technology in Tropical Diseases (INCT-DT), Department of Morphology, Federal University of Minas Gerais, Belo Horizonte 31270-901, Minas Gerais, Brazil; giunchetti@gmail.com; 8Laboratório de Bioativos e Nanobiotecnologia, Universidade Federal de São João Del-Rei, Divinópolis 35501-296, Minas Gerais, Brazil; marianacamposdapaz@ufsj.edu.br

**Keywords:** *Rickettsia rickettsii*, Rocky Mountain spotted fever, Brazilian spotted fever, diagnosis

## Abstract

Rocky Mountain or Brazilian spotted fever, caused by *Rickettsia rickettsii*, is a fulminant, seasonal, and neglected disease that occurs in focal points of North America and South America. Its rapid detection is essential for the better prognosis and survival rate of infected individuals. However, disease diagnosis still faces challenges as the accuracy of many of the available laboratory tests fluctuates. This review aimed to analyze methods for antibody or antigen detection, their gaps, and their evolution over time. A search was conducted to find all studies in the Pubmed database that described the antibody or antigen detection of *R. rickettsii* infections. Initially, a total of 403 articles were screened. Of these articles, only 17 fulfilled the pre-established inclusion criteria and were selected. Among the different methods applied, the IFA technique was the one most frequently found in the studies. However, it presented varied results such as a low specificity when using the indirect method. Other techniques, such as ELISA and immunohistochemistry, were also found, although in smaller numbers and with their own limitations. Although some studies showed promising results, there is a pressing need to find new techniques to develop a rapid and effective diagnosis of *R. rickettssi* infection.

## 1. Introduction

*Rickettsia rickettsii* [1], an important infectious agent that is part of the rickettsiosis group, is an intracellular Gram-negative coccobacillus bacterium belonging to the phylum Proteobacteria, class Alphaproteobacteria, order Rickettsiales, family Rickettsiaceae, and genus *Rickettsia*. The disease caused by this bacterium is known as Rocky Mountain spotted fever (RMSF) or Brazilian spotted fever (BSF), depending on its geographical location [2,3]. RMSF was first described by Wood in 1896 when he reported clinical data suggesting “spotted fever” as a distinct disease of unknown origin [4]. A hyperendemic outbreak in Montana’s Bitterroot Valley in the late 19th and early 20th centuries triggered more interest and research, giving rise to the name Rocky Mountain spotted fever [5,6]. Subsequently, cases were recorded in other regions of the United States and throughout the Americas, in countries such as Colombia, Brazil, Mexico, Costa Rica, Argentina, and Panama [7].

*R. rickettsii* is transmitted through tick bites and, due to its extensive distribution throughout the Americas, each region has a different species of tick as the main vector. In North America, transmission occurs through parasitiformes of the Ixodidae family, mainly by the species *Dermacentor variabilis* [8], and *Dermacentor andersoni* [9]. In South America, especially in Brazil, *Amblyomma sculptum* [10] is considered the most important *R. rickettsii* transmission vector [11,12]. In addition to these main species, other species have also been described as vectors capable of transmitting the pathogen [13,14,15]. Wild and domestic animals, such as capybaras, horses, and dogs, play an important role in the disease’s epidemiological chain, since they are the main reservoir of spotted fever transmitters, with humans as incidental hosts [16,17]. In addition, animals can also be susceptible to infection. For instance, dogs exhibit fever, lethargy, anorexia, anemia, thrombocytopenia, and potential vestibular dysfunction when exposed to the disease [18].

Symptoms usually begin on the 1st to 4th day after contact with the infected vector. During this period, the infected subject may experience fever, headaches, and photophobia, along with other milder symptoms. Rashes occur in approximately 85% of infected individuals, usually occurring on the 2nd to 4th day and spreading across the body from the wrists and ankles [19]. In more severe cases, interstitial pneumonia, meningoencephalitis, acute kidney injury, and multiple organ failure may occur after the 5th day [20]. In cases where the symptoms worsen and there is no adequate treatment, the condition can be fatal [21].

Despite being essential for a better prognosis, diagnosing the infection still has its challenges. In the acute phase, nonspecific symptoms and low bacteremia are obstacles to an accurate laboratory diagnosis [22]. Moreover, infected individuals rarely display measurable IgM antibodies in the early stages, while the presence of IgG antibodies is less specific in the late stages, as they can remain detectable in the blood for months or even years [23]. In this sense, its diagnosis mainly depends on a clinical evaluation and epidemiological background. Molecular and serological laboratory tests are applied to diagnose the infection, with the ideal test depending on the stage of the disease and the type of sample available for testing [24].

The molecular detection of rickettsial DNA by polymerase chain reaction (PCR) has become widely used for the accurate confirmation of rickettsial infections. However, the PCR technique has some limitations [25], and, due to the need for advanced resources, the use of PCR in some endemic environments is limited to reference and research laboratories [23]. Furthermore, the test’s sensitivity will depend on the sample type and the stage of infection at which the sample was collected [22]. Regarding serological methods, indirect immunofluorescence antibody (IFA) assays using paired acute and convalescent sera are the reference standard for the serological confirmation of rickettsial infection [26]. This method is superior to other techniques, such as complement fixation (CF), the latex agglutination test (LAT), and the Weil–Felix test. However, IFAs can also face problems in correctly diagnosing the disease [23,27]. The enzyme-linked immunosorbent assay (ELISA) can be used to overcome these limitations since it shows reliable performance when diagnosing various infections. In addition to methods aimed at detecting antibodies, others are used for detecting the antigens in samples, such as direct IFAs and immunohistochemistry.

Nonetheless, the detection of *R. rickettsii* remains a challenge, requiring improved laboratory diagnostic techniques to offer the best prognosis. This review set out to analyze methods for detecting the antigens or antibodies of RMSF and BSF infections and then monitoring their progress. Identifying gaps in the current knowledge can help guide new research.

## 2. Materials and Methods

For this narrative review, the search for scientific articles was conducted using the PubMed database. The search included all papers published to date. The descriptors included the boolean operators “AND” and “*”, as follows: (Rocky Mountain spotted fever[Title/Abstract]) AND (diagno*[Title/Abstract]); (Brazilian spotted fever[Title/Abstract]) AND (diagno*[Title/Abstract]); (*Rickettsia rickettsii* [Title/Abstract]) AND (diagno*[Title/Abstract]). The selected articles were screened using inclusion and exclusion criteria. Only those using antigen or antibody detection methods for diagnosing RMSF or BSF were selected. Bibliographical reviews, case studies, epidemiological reviews, molecular diagnoses and serological diagnoses of other *Rickettsia* spp., editorials, and duplicate articles were excluded. Figure 1 represents the flowchart for selecting articles.

## 3. Antigen and Antibody Detection for Diagnosing RMSF and BSF Infections

### 3.1. Serological Tests Used to Detect RMSF and BSF

Serological methods offer several advantages compared to other diagnostic methods, such as the use of noninvasive sampling, in which the necessary material can be obtained with minimal discomfort to the individual, which is less aggressive than methods that require tissue biopsies [28]. Moreover, these tests are often highly specific and can detect low antibody or antigen concentrations, making them useful for the early detection of diseases or immune responses [29]. Serological tests are generally cheaper than molecular methods, making them accessible for large-scale screening or in resource-limited settings [29].

Many serological tests can be quickly performed and do not require specialized equipment or highly trained personnel, facilitating their use in different healthcare settings. In addition, these tests can be easily scaled up and are particularly useful in epidemiological surveys or for screening blood donations. Moreover, they can be used to monitor the prevalence of diseases in a population or assess levels of immunity against certain infections, such as after vaccination campaigns [30]. However, serological methods also have limitations, such as the window period, during which antibodies may not be detectable after the initial infection; potential cross-reactivity leading to false positives; and an inability to provide information about the presence of live pathogens. It is also essential to consider the context in which these methods are used, as their performance may vary depending on the disease, the stage of infection, and the population tested [30]. Among the available serological tests, the indirect IFA is the standard one used for diagnosing RMSF and BSF [24,31]. However, other serological tests, such as the ELISA, are also used. 

#### 3.1.1. ELISA

The ELISA is a widely used diagnostic and laboratory tool [32] allowing for the detection and quantification of antigens or antibodies and used to guide the diagnosis and monitoring of diseases [33]. ELISAs can be highly sensitive, detecting very low antigen or antibody levels in a sample. Moreover, due to the use of antibodies that specifically bind to target antigens, ELISA tests can present high specificity for the substance they were designed to detect [34]. In this sense, ELISAs are a valuable tool for diagnosing a wide range of diseases. However, they are still rarely applied for *R. rickettsii* infection diagnoses.

Radulovic et al. (1993) [35] tested an immunodominant B cell epitope of *R. rickettsii* in a human humoral immune response to RMSF. In their study, an epitope-blocking ELISA (DEB-EIA) used an antigen from *R. rickettsii*. A serological panel of 35 positive serum samples was used. In addition, 50 serum samples of Mediterranean spotted fever, collected in different countries, 8 serum samples from individuals with endemic typhus, 16 serum samples from individuals with Q fever pneumonia, and 477 serum samples from individuals living beside the Adriatic Sea, where RMSF does not occur, were also used. The DEB-EIA results showed that only RMSF-positive serum samples were able to perform epitope blocking, resulting in 100% sensitivity and specificity.

#### 3.1.2. Indirect Immunofluorescence Assay

The indirect IFA is an advanced technique used to identify specific antibodies in samples through the use of two antibodies: an unlabeled primary and a secondary conjugated to a fluorophore. The primary antibody specifically binds to the target antigen, while the secondary antibody, specific to the primary antibody, is responsible for the fluorescence [26]. This indirect method has the advantage of being widely used due to its high sensitivity, signal amplification, and ability to detect multiple targets in the same sample [36].

McQuiston et al. (2014) [37] analyzed data from a surveillance program in Tennessee, USA, in which *R. rickettsii* was detected in 77% of individuals using indirect IFAs. In this work, 13 volunteers underwent visits during the acute phase of the disease, 0 to 2 weeks after the onset of symptoms, and serum was collected for IFA tests. Serum was also collected at 2 to 4 weeks, 4 to 8 weeks, and one year after symptom onset. IFA tests for IgG and IgM were performed according to the standard method, using a specific antibody for *R. rickettsii*. Based on the serological results, recent infection could not be confirmed in any infected individual, but anti-*R. rickettsii* IgM and/or IgG were present in at least one serum collection in 77% of the volunteers, occurring in 10 of 13 individuals. Anti-*R. rickettsii* IgM antibodies were commonly observed, occurring in 9 of 13 RMSF-infected individuals. However, the pattern of reactivity among them was not as expected. The first collections did not demonstrate the development of IgG, while, when present, IgM was often elevated during the first collection and did not increase in the first few weeks of infection. The last collection, one year later, showed IgM antibodies in 3 of 10 infected individuals, while IgG was present in 5 of 13. Twenty-three percent of people (3 of 13) showed both IgM and IgG antibodies one year later. These results reveal the possibility of false positives when using IgM.

Straily et al. (2020) [38] compared the level of antibody reactivity among healthy individuals in two regions of the United States and assessed the impact of the prevalence of antibodies against *R. rickettsii* on public health surveillance in one of these regions. Blood donations were collected between May and July of 2016, a time of year that coincides with the peak of reported cases. The samples were evaluated using an indirect IFA aimed at identifying the presence of IgG. As a result, 11.1% (166/1493) of donors from Georgia and 6.3% (95/1511) of donors from Oregon and Washington demonstrated reactive antibody titers, at titers ≥ 64. Moreover, only 3.1% (93/3004) of all donors had titers ≥ 128. The results suggest that a single IgG antibody titer is an unreliable measure for RMSF’s diagnosis.

#### 3.1.3. Complement Fixation

The CF technique was first described in 1901 [39,40] and is an immunological method used to detect the presence of antigens or antibodies in a biological sample [41]. The technique involves adding serum containing antibodies or antigens to a mixture of complement and antigen or antibody. The presence of an antigen or antibody in the sample results in complement fixation, which can be detected through a hemolysis reaction [41]. The quantitative CF technique is a variation that allows for quantifying the amount of antigen or antibody present in a sample [42], and its accuracy depends on the use of a satisfactory antigen and the ideal amount of complement [43,44]. This technique can be used to diagnose different types of disease-causing microorganisms, including viruses, such as influenza [45], and herpes [46]; bacteria, such as *R. rickettsii* [47] and *Treponema pallidum* [48]; and protozoa, such as *Leishmania* spp. [49] and *Trypanosoma cruzi* [50].

Lundgren, Thorpe, and Haskell (1966) [51] studied the development and persistence of CF antibodies and rickettsiosis in 183 birds of different species, from chickens to falcons, inoculated with *R. rickettsii*. Their results showed that 10 birds were positive and 173 birds were negative for *R. rickettsii*. The authors concluded that certain bird species may contribute to the spread of the etiological agent of RMSF in nature and that CF may be an inappropriate test for the serological diagnosis of *R. rickettsii* in birds.

Shepard et al. (1976) [52] tested sera from 137 individuals suspected of rickettsial infection, including RMSF, using CF. One hundred and two infected individuals were identified, of which only those whose antibody titers were equal to or greater than 1:16 were included. Among the infected individuals, only 9.5% showed maximal titer detection, while 50% had a titer detection greater than 256. The main observation in this study was the slow appearance of antibodies in the CF test for the diagnosis of *Rickettsia*. Therefore, other tests may be more efficient in terms of the rapid detection of IgM or other antibodies.

### 3.2. Direct Immunofluorescence Assay

The direct IFA is a technique characterized by the immunohistochemical detection of rickettsiae in skin biopsy samples taken from bedsores or exanthematic lesions and is considered to be a reliable assay for diagnosis [53]. The best sample and evolution time of the lesions for carrying out the examination varies. Furthermore, the procedure should cause minimal trauma to the sample, which is generally 4 mm long and involves both the epidermis and dermis [54]. This technique consists of fixing a biopsy tissue in formalin and embedding it in paraffin, before using it to detect the antigens directly in the sample. The direct technique involves the use of primary antibodies directly labeled with fluorophores, allowing the rapid identification of the antigen. This method has the advantage of being faster compared to the indirect IFA [36].

Hall and Bagley (1978) [55] described the IFA by examining fixed tissue sections rather than unfixed tissue. A modification of a trypsin digestion procedure for fixed tissues was made. In their study, rhesus monkeys, guinea pigs, and chicken embryo yolk sacs were experimentally infected, and their tissues were then fixed in formalin and embedded in paraffin. The results showed that it was possible to identify *R. rickettsii* by an IFA through both methods. Moreover, the staining intensity using the formalin-fixed method was slightly decreased compared to that in fresh tissues. Conversely, there was a notable improvement in the morphology of the formalin-fixed and paraffin-embedded tissues used in this procedure, facilitating the identification of cells and tissues that harbor rickettsiae.

A study conducted by Walker and Cain (1978) [56] investigated an IFA for specific RMSF diagnoses using fixed tissues embedded in paraffin. Autopsy samples from 10 probable cases of RMSF were analyzed. The results showed that structures with the size and shape of rickettsiae were specifically stained in the endothelium and vascular walls of renal capillaries, veins, and arteries in the kidneys of 7 out of 10 probable RMSF cases. However, the time between treatment and tissue collection may have affected the results.

Fleisher, Lennette, and Honig (1979) [57] performed an IFA to diagnose RMSF by identifying *R. rickettsii* in skin biopsy tissue from two individuals, aged 3 and 6 years, with suspected disease. Tissue staining was achieved with rabbit anti-*R. rickettsii* globulin labeled with fluorescein isothiocyanate. After all preparation steps had been completed, fluorescent microscopy was performed. Tissues from the 3-year-old individual showed, supposedly, its identification in all sections and areas of moderate fluorescence, with bright green coloration and a coccobacillary morphology visible in several locations. However, no sections from the 6-year-old individual exhibited fluorescence.

Davidson et al. (1989) [58] described the animal diagnosis of *R. rickettsii*. In their study, 14 laboratory beagles without reactive antibodies for *R. rickettsii* were used. Among them, nine dogs were inoculated intradermally with *R. rickettsii* (Shelia Smith strain), while five dogs served as controls and were similarly inoculated with an equal volume of diluent. Tissue samples were obtained before inoculation and post-inoculation on days 3, 6, 9, 12, 15, 17, and 19. Their results suggest that unaffected cutaneous inguinal skin is inferior to affected cutaneous lesions in detecting rickettsial antigens in tissues from infected dogs [55].

Melles, Colombo, and Lemos (1999) [59] evaluated the presence of *R. rickettsii* through direct IFAs in different cultures grown in concentrations of 3% and 5% of bovine serum. For this purpose, a standard sample of the *R. rickettsii* Sheyla Smith strain, cultivated in the Vero cell line, a petechial lesion sample from a patient with suspected BSF, and capybara and *Amblyomma cooperi* ticks were used. Positive results were determined by observing green, fluorescent, rickettsia-like microorganisms in the cells, demonstrating that, when using a higher percentage of fetal bovine serum for culture, such as 5%, the technique becomes more sensitive compared to adding 3% fetal bovine serum.

### 3.3. Immunohistochemistry

Immunohistochemical staining (IHC) is the set of methods in which antibodies are used as specific reagents capable of identifying and establishing a connection with the tissue constituents that function as antigens. These methods detect and localize the protein of interest within a tissue section using a specific antibody that will be detected by an enzymatic reaction (such as peroxidase or alkaline phosphatase), generating a colored chromogenic product [60]. Co-marking can also be performed using this technique, which is a qualitative or semi-quantitative method. This connection enables the location and identification of the presence of various substances in cells and tissues through the color associated with the antigen–antibody complexes formed in the meantime [61]. IHC is a powerful tool for specifically binding an antibody to an antigen to detect and localize specific antigens in cells and tissues, and it is widely used in clinical diagnoses in anatomic pathology [62].

Paddock et al. (1999) [63] described 16 fatal RMSF cases between 1996 and 1997, which were serologically unconfirmed, for which a diagnosis of RMSF was established by IHC of tissues obtained at autopsy. Serum and tissue samples from individuals with a fatal disease compatible with RMSF were also tested by indirect IFAs, where no serum demonstrated IgG or IgM antibodies reactive with *R. rickettsii*. However, the IHC staining confirmed an RMSF diagnosis in all individuals. These data suggest that IHC staining is underrecognized and underutilized as a diagnostic tool, and that many, if not most, deaths caused by *R. rickettsii* in the United States are overlooked, unconfirmed, or unreported.

### 3.4. Comparative Studies

Philip et al. (1977) [64] evaluated the CF, microimmunofluorescence (micro-IF), microagglutination (MA), and hemagglutination (HA) results for the antibody detection of *R. rickettsii* using sera from 324 individuals. Their study demonstrates that a total of 30% (97/324) of people were diagnosed by micro-IF as having RMSF, 26% (85/324) were seropositive for RMSF by MA, 30% (98/324) were confirmed as having RMSF by HA, and only 13% (43/324) were confirmed as having RMSF by CF. Of the total number of positive individuals, 86% (93/108) were deemed seropositive by two or more methods, and only 37% (40/108) were deemed positive by all four. There was good agreement between the micro-IF, MA, and HA tests, but the CF test was less sensitive than the others. Only half of the individuals considered to have RMSF by micro-IF, MA, and HA were positive with CF, whereas almost all who were positive under CF were also positive on all other tests.

The Weil–Felix test is a serological agglutination method used to detect the presence of antibodies against the bacteria of the *Rickettsia* genus in serum [65]. Hechemy (1979) [66] carried out a comparative study between the Weil–Felix test and CF, considering micro-IF as a confirmatory standard for the diagnosis of RMSF. Of the 335 reactive individuals identified by the Weil–Felix test, only 21 were detected by micro-IF. Of the 21 individuals positive in the micro-IF, only three were positive under CF. The low specificity of the Weil–Felix test was observed, as well as the low sensitivity of CF. Finally, the author warned against uncritically trusting the positive results of the Weil–Felix test or the negative results of the CF test.

Walker et al. (1980) [67] explored diagnostic methods for RMSF, aiming to assess the specificity and sensitivity of various techniques, including skin biopsy IFA, HA, CF, and Weil–Felix (Proteus Ox-2 and Proteus Ox-19 agglutination). Their analysis involved 142 serum samples and 16 skin biopsies obtained from individuals with a clinical suspicion of an RMSF infection. The sensitivity of the IFA was 70% (7/10), HA 19% (3/16), CF 0% (0/4), Proteus Ox-2 18% (3/17), and Proteus Ox-19 65% (11/17). Regarding the specificity of these diagnostic tests for RMSF, and reflecting the occurrence of false-positive results, HA and the skin biopsy IFA demonstrated the best specificity results, with values of 99% and 100%, respectively.

Clements et al. (1983) [68] evaluated IgM and IgG antibody responses in RMSF infections using ELISA and IFA techniques. Initially, the IFA was used to measure specific *Rickettsia*-class immunoglobulins in the sera obtained (before and after vaccination) from volunteers using a new formalin-inactivated vaccine for RMSF. However, the tests’ low sensitivity for detecting IgM antibodies in post-vaccination and post-infection sera led to the adaptation of an ELISA for comparison. An IFA-negative healthy volunteer was used as a negative control for the ELISA, regarding the positive controls for the IgM ELISA and the IgG ELISA; paired serum samples from early convalescent (IgM antibody titer by IFA test, 1:80) and late convalescent patients were used as reference standards. Overall, the IgM ELISA and IFA IgM test results agreed in only 13 (52%) of the 25 paired sera, while the IgG ELISA and the IFA test for total immunoglobulins gave concordant results in 85 (84%) of 101 paired sera and the concordance rate of the IgG ELISA and the IFA test for IgG was 19 of 25 paired sera. The authors stated that the ELISA test and the IFA test were uniformly specific, but that the ELISA was more sensitive than the IFA test for detecting low levels of antibodies present after vaccination and during late convalescence.

Hechemy et al. (1983) [69] conducted a comparative study on the results of latex-*R. rickettsii* tests and micro-IF for detecting antibodies against RMSF in 11 laboratories across nine U.S. states where the disease is endemic. The study was conducted during the 1980 RMSF season and used a double-blind study design, dividing the laboratories into two groups: the New York State Laboratory (NYL) and Collaborating Laboratories (CLs). The authors considered serum samples positive for *R. rickettsii* using the micro-IF method as the standard for a positive result. The latex-*R. rickettsii* test successfully diagnosed active RMSF at high titers in a single serum from individuals with an active infection. However, the test failed to detect antibodies in individuals with a previous infection. When compared with the micro-IF, the results of the latex-*R. rickettsii* method demonstrated a sensitivity of 84.45% for tests conducted in the NYL and 79.20% for tests conducted in the CLs. Moreover, the efficiency of the latex-*R. rickettsii* test was 96.79% for the NYL and 93.30% for the CLs. Both tests were capable of detecting antibodies one week after the onset of infection. However, with micro-IF, titers appeared to remain above the minimum significant reactivity levels for *R. rickettsii* in comparison to the latex-*R. Rickettsii* titers.

White, Patrick, and Miller (1994) [70] evaluated 23 individuals with suspected RMSF using direct IFAs and immunoperoxidase tests. During tissue preparation, part of the tissues was frozen for IFAs and the other part was fixed in formalin, routinely processed, and embedded in paraffin for staining with immunoperoxidase, hematoxylin, and eosin. Ten of the twenty-three individuals were identified as having RMSF, of which nine were positive for both tests. The use of immunoperoxidase on paraffin-embedded tissue provided results essentially identical to the direct immunofluorescence of fresh-frozen biopsy material. In the only case that presented a false negative, the individual had received anti-rickettsial antibiotics 72 h before the biopsy. Therefore, no significant differences were observed between the two methods. Table 1 summarizes the main points of the above-mentioned studies. 

## 4. Discussion

Although it is not a commonly reported disease, the mortality rate of *R. rickettsii* infections can reach 20% to 30% without early treatment. The proper diagnosis of the disease may be hampered by the fact that it presents nonspecific symptoms and can be misdiagnosed or confused with other diseases, such as dengue, malaria, or ehrlichiosis [71]. Moreover, the recommended laboratory diagnosis still has its negatives, which act as a barrier to an accurate diagnosis. Due to these bottlenecks, there is a substantial need to develop accurate and specific methods capable of diagnosing the disease and identifying the best treatment for the infected individual [72]. The scarcity of research in this field may be explained by the socioeconomic profile of the affected populations, which often reside in low-income areas. These populations are more exposed to tick vectors due to problematic living and working conditions, in addition to precarious housing and proximity to habitats that attract the vectors, thereby elevating their risk of contact with ticks [73].

Despite the scarcity of research, especially when compared to studies conducted on other diseases, over the last years several researchers have conducted studies employing different techniques for diagnosing this infection, including the indirect IFA, which is still considered the gold standard serological method for detecting RMSF and BSF infections. From the above-mentioned results, it is possible to observe that studies that used both direct and indirect IFAs have demonstrated good results in diagnosing the disease. However, the sensitivity and specificity achieved using these techniques is still far from ideal. Other techniques, such as immunohistochemistry and ELISA, were also successfully used, demonstrating the ability to distinguish between negative and positive serum samples. However, since only a limited number of studies have used these techniques, it is not possible to infer whether they are reliable when it comes to diagnosing RMSF or BSF infections. Furthermore, other methods, such as the CF and Weil–Felix techniques, did not demonstrate good diagnostic performance, with both having low detection results in the above-mentioned studies. Indeed, the studies that compared different diagnostic tests showed that CF was one of the least promising techniques, followed by the Weil–Felix. Furthermore, in comparative studies, the IFA demonstrated the greatest diagnostic capacity.

Despite the good results seen in some studies, the diagnostic scenario is still far from the ideal. Studies that employed the indirect IFA demonstrated varied sensitivity and specificity values. This is worrisome, since this method is the recommended assay for antibody detection for disease diagnosis. Therefore, the need to re-evaluate current diagnostic protocols is of the utmost importance. In fact, the serological diagnosis of the disease still faces several challenges, mainly due to the fact that when infected individuals seek medical assistance, their antibody levels are generally undetectable by the recommended method, generating erroneous results. Furthermore, when using serum samples, the ideal diagnosis should be made by comparing sera from the acute and convalescent phases, which is not always possible [23]. In this sense, the fact that not all of the studies mentioned that employed sera as samples yielded results consistent with paired sera may compromise the reliability of these results, as it is not ideal to use only sera from the acute phase to diagnose the disease. In addition, the variety in the diagnostic capacity of the other methods, such as direct IFA, is notable, demonstrating the flaws in these methods as well. Moreover, the scarcity of recent studies is a source of concern. Only two studies were published in this century, suggesting a lack of interest in developing an improved diagnostic method. This lack of study cannot be justified by a lack of case notifications, since the number of cases is multiplying in some regions. In Brazil, for example, there was an increase in case notifications from 2021 to 2022, confirming that the infection is present, and the rise in cases is worrisome [74]. In this sense, it is hypothesized that the lack of studies reflects the diagnostic scenario in recent decades, since the true number of disease cases may be underestimated due to inefficient diagnosis. Furthermore, few studies are focusing on the adaption of current technologies or the development of new antigens to improve existing techniques; this underscores the need to develop a more effective diagnosis for this disease. 

Some measures to improve diagnostic performance can be pointed out, such as investing in faster methods that are easier to store and manage, in addition to presenting simplified testing, such as point-of-care platforms. Moreover, there is an urgent need to develop new studies aiming at developing and testing new antigens for serological tests, such as recombinant proteins, multiepitope proteins, and synthetic peptides, which represent a promising strategy to boost the sensitivity and specificity of serological tests. In fact, improving the sensitivity of these tests could reflect positively on early disease diagnosis. The antigens mentioned above offer the advantage of functioning without the need for specific biosafety requirements when handling microorganisms, as well as being more suitable for assay standardization [75]. Moreover, investing in the use of different methods is another path, such as electrochemical immunosensors, which provide a more efficient option for detecting reactive antibodies to *R. rickettsii* with high specificity and sensitivity, thus reducing the sample volume requiring analysis [76]. Also, due to the coexistence of more than one pathogen that can be transmitted by the same vector, it is essential to develop a serological test capable of detecting tick-borne diseases, mainly for screening cases. For example, a recent study detected the co-occurrence of *Borrelia* spp. and *Rickettsia* spp. in *A. sculptum* vectors in Brazil [77], which underlines the importance of using multiple tests. Finally, studies on the development of new tests could be improved, such as expanding the sample size supported by statistical programs capable of arriving at more reliable results.

In summary, the fact that RMSF and BSF diagnoses based on antigen or antibody detection encounter significant bottlenecks prompts the need for the development of a better diagnostic method. In fact, it was possible to observe a notable variance in the diagnostic capacity of the studies mentioned above, making the correct diagnosis of the disease a challenge. This scenario is an obstacle to the correct treatment of the disease, impacting the prognosis of infected individuals, as well as being an obstacle to controlling the spread of the infectious agent. In the light of these observations, it is essential to improve diagnostic methods to obtain a quick and effective diagnosis, with the aim of facilitating early detection and pathogen dispersion and assisting in the creation of more efficient prevention strategies.

## Figures and Tables

**Figure 1 biomedicines-12-01501-f001:**
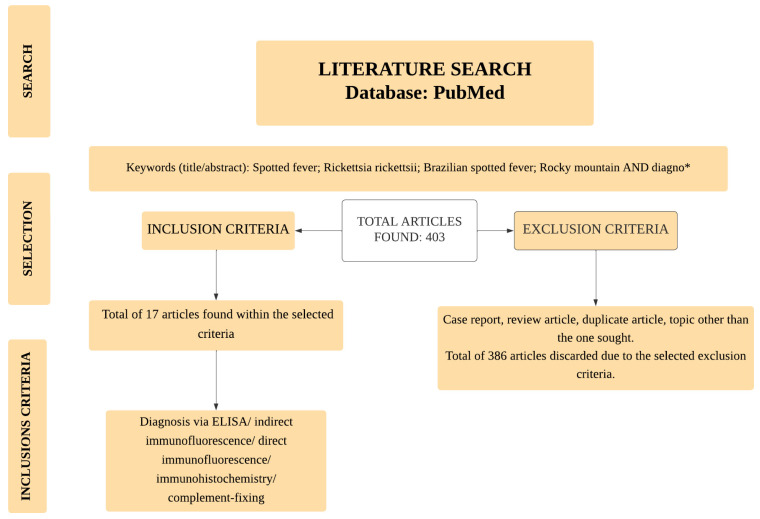
Article flowchart selection. *: boolean operator.

**Table 1 biomedicines-12-01501-t001:** Antibody and antigen detection studies for RMSF and BSF infection diagnoses. CF: complement fixation; IFA: immunofluorescence assay; IHC: immunohistochemical staining; HA: hemagglutination; MA: microagglutination; Micro-IF: microimmunofluorescence.

Disease Stage	Infected Host	Sample Used	Method	Results	Author/Country
Acute	Animal	Serum	CF	CF was not able to detect antibodies from infected chickens, pheasants, sparrow hawks, magpies, or ravens CF detected antibodies from infected pigeons with maximal detection between the 3rd and 5th infection week CF detected antibodies from one red-tailed hawk and one marsh hawk in the 2nd and the 3rd infection week	Lundgren, Thorpe, and Haskell (1966)/USA [51]
-	Human	Serum	CF	9.5% of infected individuals showed maximal titer detection 64% of serum samples from infected individuals showed cross-reaction with typhus antigens	Shepard et al., (1976)/USA [52]
Acute and convalescent-phase	Human	Serum	Micro-IF MA HA CF	Reactivity - Micro-IF: 30% MA: 26% HA: 30% CF: 13%	Philip et. al. (1977)/USA [64]
Convalescent-phase	Animal	Kidney and skin	Direct IFA	Direct IFA detected 7 (7/10) samples from infected individuals Samples from individuals negative for RMSF did not show immunofluorescence	Walker and Cain (1978)/USA [56]
-	Human and animal	Lung, heart, epididymis, and testis	Direct IFA	The formalin-fixed method showed a slightly reduced intensity of staining, with improved morphology Normal tissues showed no staining	Hall and Bagley (1978)/USA [55]
-	Human	Serum	Weil–Felix Micro-IF CF	Weil–Felix: only 6% of similarity with micro-IF. Agreement between these two tests was higher when considering paired serums CF: 86% false-negative	Hechemy (1979)/USA [66]
Acute	Human	Skin	Direct IFA	Direct IFA was able to detect the presence of *R. rickettsii* in only 1 (1/2) positive sample	Fleisher, Lennette, and Honig (1979)/USA [57]
Acute	Human	Serum and skin	Direct IFA CF HA Weil–Felix	Direct IFA - Sensitivity: 70% Specificity: 100% CF - Sensitivity: 0% Specificity: 0% HA - Sensitivity: 19% Specificity: 99% Weil–Felix Proteus Ox-2 - Sensitivity: 18% Specificity: 96% Weil–Felix Proteus Ox-19 - Sensitivity: 65% Specificity: 78%	Walker et. al. (1980)/USA [67]
Acute and convalescent-phase	Human	Serum	ELISA and Indirect IFA	ELISA - Sensitivity: 100% Specificity: 100% IFA - Sensitivity: 100% Specificity: 83%	Clements et. al. (1983)/USA [68]
-	Human	Serum	Latex-*R. rickettsii* test and micro-IF	New York State laboratory - Sensitivity: 84.45% Specificity: 99.98% Collaborating laboratories - Sensitivity: 79.20% Specificity: 95.81%	Hechemy et. al. (1983)/USA [69]
-	Animal	Skin	Direct IFA	Direct IFA was able to detect the presence of *R. rickettsii* in 18 (18/23) samples from erythematous macules Using normal inguinal skin, direct IFA was unable to detect the presence of *R. rickettsii*	Davidson et. al, (1989)/USA [58]
-	Human	Serum	ELISA	Sensitivity: 100% Specificity: 100%	Radulovic et. al. (1993)/USA [35]
-	Human	Serum Skin biopsy, and autopsy samples	Direct IFA and immunoperoxidase test	Direct IFA and immunoperoxidase tests detected the presence of *R. rickettsii* in 9 (9/10) positive samples	White, Patrick, and Miller (1994)/USA [70]
-	Human and animal	Skin	Direct IFA	Sensitivity was improved when using a higher concentration of fetal bovine serum	Melles, Colombo, and Lemos (1999)/Brazil [59]
-	Human	Serum whole blood liver, myocardium, spleen, kidney, lung, adrenal gland, pancreas, cerebral cortex, cerebellum, skin, stomach, colon, bone marrow, lymph node, small intestine, trachea, skeletal muscle, thymus, thyroid, coronary artery, aorta, hippocampus, medulla, pons, pineal gland, choroid plexus, ovary, tongue, and appendix	IHC	IHC was able to detect the presence of *R. rickettsii* in 12 (12/16) samples	Paddock et. al. (1999)/USA [63]
Acute and convalesce- cent-phase	Human	Serum	Indirect IFA	Anti-*R. rickettsii* IgM and/or IgG were detected in at least one collected serum sample from 10 (10/13) infected individuals	McQuiston et. al. (2014)/USA [37]
Acute and convalesce- cent-phase	Human	Serum	Indirect IFA	Only 11.1% of Georgia donors and 6% of Pacific Northwest donors had IgG titers ≥ 64	Straily et. al. (2020)/USA [38]

## Data Availability

Not applicable.

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
