# Peer review of "A Mini-Review of Diagnostic Methods for the Antigen and Antibody Detection of Rocky Mountain and Brazilian Spotted Fever"

_biomedicines, 2024, doi:10.3390/biomedicines12071501_

Round 1

Reviewer 1 Report

Comments and Suggestions for Authors

It causes confusion that the authors in this mini review mention that publications related to molecular diagnosis were excluded; however, their advantages and disadvantages were included in the review. Examples:

Lines 84- 85 “Molecular and serological laboratory tests are applied to diagnose the infection, with the ideal test depending on the stage of the disease and the type of sample available for testing [22]”.

Lines 89-90 “the use of PCR in some endemic environments is limited to reference and research laboratories [23]”

Line 124 “Serological tests are generally cheaper than molecular methods, making them accessible for large-scale screening or in resource-limited settings [27]”.

According to the article selection criteria in this mini review, these references should not be considered. If it is strictly necessary to include them, justify why

25. Ilkhani H, Hedayat N, Farhad S. Novel approaches for rapid detection of COVID-19 during the pandemic: A review. Anal Biochem. 2021 Dec 1;634:114362. doi: 10.1016/j.ab.2021.114362.

26. Gong F, Wei HX, Li Q, Liu L, Li B. Evaluation and Comparison of Serological Methods for COVID-19 Diagnosis. Front Mol Biosci. 2021 Jul 23;8:682405. doi: 10.3389/fmolb.2021.682405.

Author Response

All changes made to the manuscript are highlighted in green. Response to the Reviewers' comments are in blue.

It causes confusion that the authors in this mini review mention that publications related to molecular diagnosis were excluded; however, their advantages and disadvantages were included in the review. Examples:

Lines 84- 85 “Molecular and serological laboratory tests are applied to diagnose the infection, with the ideal test depending on the stage of the disease and the type of sample available for testing [22]”.

Lines 89-90 “the use of PCR in some endemic environments is limited to reference and research laboratories [23]”

Line 124 “Serological tests are generally cheaper than molecular methods, making them accessible for large-scale screening or in resource-limited settings [27]”.

Response - We appreciate this comment. Although studies employing molecular tests was an exclusion criterion for the selection of articles, they are an option for diagnosing the disease and we mention these tests throughout the text to show, mainly, that molecular methods have disadvantages and to improve the diagnosis of the disease, mainly aiming at better cost-benefit, serological tests are the best option. Therefore, we mention the use of molecular tests only to highlight the advantages of using serological tests.

According to the article selection criteria in this mini review, these references should not be considered. If it is strictly necessary to include them, justify why

  1. Ilkhani H, Hedayat N, Farhad S. Novel approaches for rapid detection of COVID-19 during the pandemic: A review. Anal Biochem. 2021 Dec 1;634:114362. doi: 10.1016/j.ab.2021.114362.
  2. Gong F, Wei HX, Li Q, Liu L, Li B. Evaluation and Comparison of Serological Methods for COVID-19 Diagnosis. Front Mol Biosci. 2021 Jul 23;8:682405. doi: 10.3389/fmolb.2021.68240

Response - We appreciate this comment. The reference number 25 and 26 were replaced in the text with the following references:

- Vaca, D. J., Dobler, G., Fischer, S. F., Keller, C., Konrad, M., von Loewenich, F. D., Orenga, S., Sapre, S. U., van Belkum, A., & Kempf, V. A. J. Contemporary diagnostics for medically relevant fastidious microorganisms belonging to the genera Anaplasma,Bartonella,Coxiella,Orientia and Rickettsia. FEMS microbiology reviews, 2022, 46(4), 10.1093/femsre/fuac013.

- Bharadwaj, M., Bengtson, M., Golverdingen, M., Waling, L., & Dekker, C. Diagnosing point-of-care diagnostics for neglected tropical diseases. PLoS neglected tropical diseases, 2021, 15(6), e0009405, 10.1371/journal.pntd.0009405.

Reviewer 2 Report

Comments and Suggestions for Authors

Rocky Mountain spotted fever (RMSF) or Brazilian spotted fever (BSF) are tick-borne infections that are on the rise especially in the Americas. In undiagnosed and untreated forms, mortality is 20-30%. For this reason it is important that the diagnosis is carried out early, and this fact can cause difficulties both due to the clinical picture sometimes with non-specific symptoms, and because laboratory tests can in some cases give false negatives.

A review of various laboratory methods with indirect techniques as ELISA, IFI, Complement Fixation, Micro-IF, cryoagglutination (MA) and haemagglutination (HA), as well as immunohistochemistry and direct tests (PCR) for the diagnosis of RMSF / BSF is carried out.

To improve laboratory performance, the Authors highlight the importance of developing new antigens for serological tests, such as recombinant proteins and synthetic peptides, which represent a promising strategy to increase the sensitivity and specificity of serological tests. Furthermore, these antigens offer the advantage that they do not have biosafety requirements, which are essential in the manipulation of microorganisms.

The work is done well and precisely, however when you are bitten by a tick, even in areas where the presence of RMSF / BSF is known, this is not always the disease transmitted. When Willy Burgdorfer collected Ixodidae ticks on Long Island to study Rickettsia rickettii, he found himself analyzing ticks with Borrelia burgdorferi. For this reason I would dedicate a few lines to this and the utility of multiplex testing for tick-bites.

The work can be accepted with these small modifications.

Author Response

All changes made to the manuscript are highlighted in green. Response to the Reviewers' comments are in blue.

Rocky Mountain spotted fever (RMSF) or Brazilian spotted fever (BSF) are tick-borne infections that are on the rise especially in the Americas. In undiagnosed and untreated forms, mortality is 20-30%. For this reason it is important that the diagnosis is carried out early, and this fact can cause difficulties both due to the clinical picture sometimes with non-specific symptoms, and because laboratory tests can in some cases give false negatives.

A review of various laboratory methods with indirect techniques as ELISA, IFI, Complement Fixation, Micro-IF, cryoagglutination (MA) and haemagglutination (HA), as well as immunohistochemistry and direct tests (PCR) for the diagnosis of RMSF / BSF is carried out.

To improve laboratory performance, the Authors highlight the importance of developing new antigens for serological tests, such as recombinant proteins and synthetic peptides, which represent a promising strategy to increase the sensitivity and specificity of serological tests. Furthermore, these antigens offer the advantage that they do not have biosafety requirements, which are essential in the manipulation of microorganisms.

The work is done well and precisely, however when you are bitten by a tick, even in areas where the presence of RMSF / BSF is known, this is not always the disease transmitted. When Willy Burgdorfer collected Ixodidae ticks on Long Island to study Rickettsia rickettii, he found himself analyzing ticks with Borrelia burgdorferi. For this reason I would dedicate a few lines to this and the utility of multiplex testing for tick-bites.

The work can be accepted with these small modifications.

Response - Dear reviewer, thank you for your contribution. As suggested, we have added a small discussion about this subject, as follows:

“Also, due to the co-existence of more than one pathogen that can be transmitted by a vector from the same family or even by the same vector, it is essential to develop a serological test capable of detecting tick-borne diseases, mainly for screening cases. For example, a recent study detected the cooccurrence of Borrelia spp. and Rickettsia spp. in A. sculptum vectors in Brazil [77], highlighting the importance of multiple testing.”

Reviewer 3 Report

Comments and Suggestions for Authors

In the reviewed MS, the authors discuss various methods for antigen and antibody detection of Rickettsia pathogens, causing serious diseases such as Rocky Mountain spotted fever or Brazilian spotted fever. The authors focuses on immunological methods and provide their comparison based on literature data. It should be mentioned that there are some other papers published in the recent years on the same topic in different journals. The authors are requested to specify what exactly new their study reveal in comparison to those studies (e.g. 10.3390/pathogens10101319, 10.1016/j.actatropica.2021.105887, 10.4269/ajtmh.21-0757 and others). Additionally, to avoid repetition, the authors could summarize their results in a Table comparing advantages and disadvantages of various methods they discuss in the text. Finally, the Discussion is written in suboptimal way and needs revision. Some additional remarks are below.

36: Please, consider removing “Therefore” and change “identify” to “compare/analyze”

39: 17 or 19? There are 19 according to Fig. 1

50: Please, provide brief taxonomy of this bacteria as it is usually done

60: please specify (Parasitifromes: Ixodida) for “ticks”

63-64: please, specify whether Amblyomma and Dermacentor are the only two ixodid genera transmitting this pathogen or not.

102-103: “This review set out to identify methods for detecting antigens or antibodies of RMSF and BSF infection and then monitor their progress.” In the paragraph above, you have already “identified” these methods. Please, reword the aim of your review.

113-115: “Bibliographical reviews, case studies, epidemiological reviews, molecular diagnosis and serological diagnosis of other diseases, editorials, duplicate articles, and articles related to other subjects were excluded” --- this is a bit unclear. Some references from the reference list (e.g. [20] and some others) quite meet the listed criteria.

249, 257 (check all over the text): R. rickettsia --- italic

357-365: this text corresponds to Introduction, not Discussion. Please, consider transferring it to Introduction.

449 – Dermacentor - italic

Table 1: check the caption (it should not be in capital letters). The title of the Table is suboptimal. I did not find the reference to the Table 1 in the tex.

Please check all reference numbers in the Table1: e.g. Shepard et al.,(1976) / USA [49] should be [48]

The Discussion needs revision. Please, report the most important conclusions followed from your research. The last phrase of Discussion is very uncertain and unspecific.

Comments on the Quality of English Language

minor

Author Response

All changes made to the manuscript are highlighted in green. Response to the Reviewers' comments are in blue.

In the reviewed MS, the authors discuss various methods for antigen and antibody detection of Rickettsia pathogens, causing serious diseases such as Rocky Mountain spotted fever or Brazilian spotted fever. The authors focuses on immunological methods and provide their comparison based on literature data. It should be mentioned that there are some other papers published in the recent years on the same topic in different journals. The authors are requested to specify what exactly new their study reveal in comparison to those studies (e.g. 10.3390/pathogens10101319, 10.1016/j.actatropica.2021.105887, 10.4269/ajtmh.21-0757 and others). Additionally, to avoid repetition, the authors could summarize their results in a Table comparing advantages and disadvantages of various methods they discuss in the text. Finally, the Discussion is written in suboptimal way and needs revision. Some additional remarks are below.

Response - Dear reviewer, thank you for your contribution. Before starting to write this article, we checked the literature to see if there was any article already published with a similar structure and proposal to the one we intended. In fact, we were aware of the publications of the articles cited in the comment, and we even cited one of them. In this sense, our article differs from other articles published to date, as we focus specifically on serological studies, describing all articles that exist to date in the literature and detailing how each study was conducted, in addition to focusing only on the development of diagnosis for R. rickettsii. Regarding the suggested table on the advantages and disadvantages of each method, we preferred not to create this table since these points were described in the text. Furthermore, by creating this table, we would be diverting some attention from the main focus of our article, which is what has been developed over the years to improve the diagnosis of R. rickettsii. Furthermore, by creating this table, we could lead the reader to think that a certain method is better than another, and this is not the message we want to convey.

36: Please, consider removing “Therefore” and change “identify” to “compare/analyze”

Response - We appreciate this comment. As suggested, the sentence has been changed, as follows: “..the available laboratory tests fluctuates. This review aimed to analyze methods for antibody or antigen detection, its gaps, and evolution over time.”

39: 17 or 19? There are 19 according to Fig. 1

Response - We are thankful for this comment. We have corrected the error in the text, being 17 being the correct number of articles found for this review.

50: Please, provide brief taxonomy of this bacteria as it is usually done

Response - We appreciate this comment. A brief taxonomy of R. rickettsii has been added, as follows: “Rickettsia rickettsii [1], an important infectious agent that is part of the rickettsiosis group, is an intracellular gram-negative coccobacillus bacterium belonging to phylum Proteobacteria, class Alphaproteobacteria, order Rickettsiales, family Rickettsiaceae, genus Rickettsia. The disease caused by this bacterium is known as Rocky Mountain spotted fever (RMSF) or Brazilian spotted fever (BSF), depending on its geographical location [2,3].”

60: please specify (Parasitifromes: Ixodida) for “ticks”

Response - We appreciate this comment. As suggested, the information has been added, as follows:

“In North America, transmission occurs through parasitiformes of the Ixodidae family, primarily by the species Dermacentor variabilis [8], and Dermacentor andersoni [9].”

63-64: please, specify whether Amblyomma and Dermacentor are the only two ixodid genera transmitting this pathogen or not.

Response - We are thankful for this comment. A short sentence has been added to clarify the question raised, as follows:

“In addition to these main species, others species have also been described as vectors capable of transmitting the bacterium [13,14,15].”.

102-103: “This review set out to identify methods for detecting antigens or antibodies of RMSF and BSF infection and then monitor their progress.” In the paragraph above, you have already “identified” these methods. Please, reword the aim of your review.

Response - We appreciate this comment. The sentencen has been changed, as follows:

“This review set out to analyze methods for detecting antigens or antibodies of RMSF and BSF infection ...”.

113-115: “Bibliographical reviews, case studies, epidemiological reviews, molecular diagnosis and serological diagnosis of other diseases, editorials, duplicate articles, and articles related to other subjects were excluded” --- this is a bit unclear. Some references from the reference list (e.g. [20] and some others) quite meet the listed criteria.

Response - We are thankful for this comment. In fact, the criteria mentioned above only concern the selection of articles that were the main theme of this review article, that is, they are criteria for the selection of articles that applied antigen/antibody detection methods for the diagnosis of the disease, which are described in topic 3. For the introduction and discussion, these criteria were not used.

249, 257 (check all over the text): R. rickettsia --- italic

Response - We appreciate this observation. The text has been re-checked and all scientific names have been italicized.

357-365: this text corresponds to Introduction, not Discussion. Please, consider transferring it to Introduction.

Response - We appreciate this comment. We removed some parts of this first paragraph. However, we chose to keep this paragraph in the discussion, since most discussions begin with a brief overview and bottlenecks of the disease topic.

449 – Dermacentor – italic

Response - We appreciate this observation. The word has been italicized.

Table 1: check the caption (it should not be in capital letters). The title of the Table is suboptimal. I did not find the reference to the Table 1 in the tex.

Response - We are thankful for this comment. The title of the table has been changed. Moreover, the reference to the Table 1 has been added, as follows:

“…observed between the two methods [67]. Table 1 summarizes the main points of the above-mentioned studies.”

Please check all reference numbers in the Table1: e.g. Shepard et al.,(1976) / USA [49] should be [48]

Response - We appreciate this observation. All the references number have been checked.

The Discussion needs revision. Please, report the most important conclusions followed from your research. The last phrase of Discussion is very uncertain and unspecific.

Response - We appreciate this comment. As suggested, the discussion was revised and modified.

Round 2

Reviewer 3 Report

Comments and Suggestions for Authors

Trypanosoma Cruzi --- T. cruzi

455: [T]he authors --- [T] should bot be bold

Comments on the Quality of English Language

minor

Author Response

Dear Dr. Alexsandro Galdino,

Thank you for your email. We have returned your manuscript for revisions
based on the following comments from the third reviewer:

"Trypanosoma Cruzi --- T. cruzi

Response:  Corrected. Thanks

455: [T]he authors --- [T] should bot be bold"

Response: Corrected. Thanks